# Deep Multi-Modal Skin-Imaging-Based Information-Switching Network for Skin Lesion Recognition

**DOI:** 10.3390/bioengineering12030282

**Published:** 2025-03-12

**Authors:** Yingzhe Yu, Huiqiong Jia, Li Zhang, Suling Xu, Xiaoxia Zhu, Jiucun Wang, Fangfang Wang, Lianyi Han, Haoqiang Jiang, Qiongyan Zhou, Chao Xin

**Affiliations:** 1The First Affiliated Hospital of Ningbo University, Ningbo 315211, China; yuyingzhe90@foxmail.com (Y.Y.); xusuling@nbu.edu.cn (S.X.); nb_zhuxiaoxia@126.com (X.Z.); fyywangfangfang@nbu.edu.cn (F.W.); 2Department of Laboratory Medicine, The First Affiliated Hospital, Zhejiang University School of Medicine, Hangzhou 310000, China; jiahq@zju.edu.cn; 3Key Laboratory of Clinical In Vitro Diagnostic Techniques of Zhejiang Province, Hangzhou 310000, China; 4Department of Dermatology, The First Hospital of China Medical University, Shenyang 110001, China; lizhang_1001@126.com; 5State Key Laboratory of Genetic Engineering, Collaborative Innovation Center for Genetics and Development, School of Life Sciences and Human Phenome Institute, Fudan University, Shanghai 200433, China; jcwang@fudan.edu.cn; 6Greater Bay Area Institute of Precision Medicine (Guangzhou), School of Life Sciences, Fudan University, Shanghai 315211, China; hanlianyi@fudan.edu.cn (L.H.); hqjiang23@m.fudan.edu.cn (H.J.)

**Keywords:** skin lesion, deep multi-modal network, recognition, information switching

## Abstract

The rising prevalence of skin lesions places a heavy burden on global health resources and necessitates an early and precise diagnosis for successful treatment. The diagnostic potential of recent multi-modal skin lesion detection algorithms is limited because they ignore dynamic interactions and information sharing across modalities at various feature scales. To address this, we propose a deep learning framework, Multi-Modal Skin-Imaging-based Information-Switching Network (MDSIS-Net), for end-to-end skin lesion recognition. MDSIS-Net extracts intra-modality features using transfer learning in a multi-scale fully shared convolutional neural network and introduces an innovative information-switching module. A cross-attention mechanism dynamically calibrates and integrates features across modalities to improve inter-modality associations and feature representation in this module. MDSIS-Net is tested on clinical disfiguring dermatosis data and the public Derm7pt melanoma dataset. A Visually Intelligent System for Image Analysis (VISIA) captures five modalities: spots, red marks, ultraviolet (UV) spots, porphyrins, and brown spots for disfiguring dermatosis. The model performs better than existing approaches with an mAP of 0.967, accuracy of 0.960, precision of 0.935, recall of 0.960, and f1-score of 0.947. Using clinical and dermoscopic pictures from the Derm7pt dataset, MDSIS-Net outperforms current benchmarks for melanoma, with an mAP of 0.877, accuracy of 0.907, precision of 0.911, recall of 0.815, and f1-score of 0.851. The model’s interpretability is proven by Grad-CAM heatmaps correlating with clinical diagnostic focus areas. In conclusion, our deep multi-modal information-switching model enhances skin lesion identification by capturing relationship features and fine-grained details across multi-modal images, improving both accuracy and interpretability. This work advances clinical decision making and lays a foundation for future developments in skin lesion diagnosis and treatment.

## 1. Introduction

The skin, the biggest organ in the human body, performs several functions, including regulating body temperature, identifying environmental cues, and protecting interior organs. Skin illnesses include a broad spectrum of disorders that impact the skin and its supporting organs, encompassing both benign and malignant skin lesions. In particular, two skin lesions that are commonly observed are malignant melanoma (MM) and disfiguring dermatosis (DD).

DD, such as dermatitis, melasma, and acne, can lead to skin inflammation, pigmentation alterations, and texture changes, significantly impacting an individual’s physical appearance and quality of life [1,2,3,4]. MM, a highly aggressive skin cancer, is prone to metastasis and recurrence, often spreading to vital organs like the lungs, liver, and bones, causing fatal damage [5,6]. Early and accurate diagnosis is crucial for the effective management and treatment of these conditions. VISIA multi-modal imaging technology has emerged as a valuable tool for identifying, diagnosing, and evaluating the severity of DD. This cutting-edge system integrates various imaging modalities, including UV, visible, and near-infrared light, to capture high-quality skin images [7]. This technology helps dermatologists analyze skin conditions, barrier function, and pigmentation. MM can be detected early using dermoscopic and clinical imaging, but integrating and interpreting multi-modal data remains challenging [8]. DD and MM often exhibit complex skin features like UV spots, brown spots, red marks, porphyrins, uneven coloration, and irregular borders. Traditional methods may lack accuracy, and physician subjectivity can influence diagnosis. Combining advanced imaging with deep learning allows for more precise and thorough skin analysis [9]. Moreover, this combination assists clinicians in recognizing subtle variations in skin characteristics, which are crucial for the identification and diagnosis of dermatological conditions [10]. Deep learning enables multi-modal imaging systems to analyze extensive data, offering precise insights for personalized treatment plans and clinical follow-up for skin lesions.

Previous approaches for skin lesion image recognition relied on conventional image processing and computer vision techniques, focusing on manual feature extraction like color, texture, and shape using methods such as the Hough transform and edge detection [11,12,13]. These features were then classified using various classifiers such as support vector machines and k-nearest neighbors [14]. Some studies used geometric models based on morphological operations for lesions with distinct features [12]. Nonetheless, these methods were limited by their dependence on hand-crafted features and classifiers, making them prone to noise and data imbalance. The advent of deep learning techniques, particularly the rise of convolutional neural networks (CNNs), has markedly enhanced the accuracy and performance of skin disease image recognition [15,16]. CNNs can extract features from clinical and dermoscopic images, enabling the high-accuracy classification of conditions like acne, eczema, psoriasis, and skin tumors [16,17]. Pre-trained models like VGG and ResNet serve as feature extractors, reducing training time and data requirements while improving accuracy [18]. Transfer learning further enhances model performance by leveraging knowledge from other medical image tasks, addressing data scarcity, and improving generalization. Generative adversarial networks (GANs) have become a useful tool for data augmentation in the field of skin disease recognition [19]. The attention mechanism, inspired by human visual processes, enhances skin disease recognition accuracy by enabling models to focus on relevant regions, such as lesion areas, while suppressing irrelevant background noise [20,21]. This attention mechanism enables the model to focus on the most relevant regions within the skin images, such as lesion areas, while suppressing irrelevant background noise [22]. This focused attention on lesion regions improves the model’s ability to distinguish skin disorders like acne, eczema, psoriasis, and skin tumors, enhancing classification accuracy. In multi-modal recognition, deep learning and vision transformers integrate skin images with patient metadata, boosting diagnostic precision. This focused attention on lesion regions enhances diagnostic accuracy [23,24].

However, current approaches toward the early detection of skin lesions predominantly rely on the analysis of individual images [16,24]. This methodology presents constraints in the holistic examination of skin ailments. Our research circumvents this limitation by employing multi-modal imaging. Moreover, the prevailing trend in multi-modal methodologies mainly centers on the concatenation and fusion of distinct diagnostic modal features, often overlooking the intricate interplay between these modes and the critically important exchanges of multi-scale feature information [9,20,24]. To address this challenge, our study introduces a deep multi-modal information-switching network tailored for the end-to-end recognition of skin lesions. It harnesses the power of deep convolutional neural networks to elicit nuanced features specific to skin lesions across various skin image modalities. The network further augments the exchange of information between features from different modalities across multiple scales, thus ameliorating both the interpretability and recognition precision in multi-modal skin imaging for skin lesions. Figure 1 illustrates the distinction between our proposed method and traditional multi-modal classification models. Unlike conventional approaches that directly fuse the features extracted by the neural network, our method employs an information-switching module to exchange information between feature maps of different modalities at various scales. This enables the recognition model to better consider the relationships between modalities.

In summary, our study aims to leverage multi-modal images to enable a more comprehensive analysis of skin conditions and enhance the accuracy of identifying skin lesions. For this purpose, we utilize a deep learning analysis approach that integrates multi-modal influences to automatically identify and classify various skin lesions. This article proposes a deep multi-modal information-switching network, known as MDSIS-Net. MDSIS-Net introduces an innovative approach that uses transfer learning within a multi-scale fully shared convolutional neural network to extract skin lesion features across different scales and modalities. Within MDSIS-Net, a novel multi-scale information-switching structure is developed to generate learnable parameters for facilitating the exchange of deep and shallow features between different modalities. The primary contributions of this work are as follows:

1. This research proposes MDSIS-Net, an end-to-end deep multi-modal information-switching network for skin lesion recognition, utilizing deep convolutional neural networks to extract intra-modality features and enhance multi-scale inter-modality interactions, improving interpretability and accuracy for DD and MM diagnosis.

2. We propose a novel multi-scale information-switching structure within a multi-modal skin lesion recognition framework, which generates learnable parameters to facilitate cross-modal deep and shallow feature exchange with adaptive layer-wise weighting. Our MDSIS-Net improves the inter-modality association of deep and shallow features within each modality, optimizing the distribution of distinct features related to different diseases across diverse imaging modalities.

3. The proposed model is trained and validated on real-world clinical data for DD and the public Derm7pt dataset for MM, demonstrating superior performance over state-of-the-art methods. Interpretability analysis generates modality-specific heatmaps, highlighting distinctive features of DD and MM to enhance model transparency and reliability. These visualizations provide dermatologists with clear insights into lesion characteristics, facilitating more informed and effective treatment decisions.

## 2. Review of the Previous Research and Literature

Traditional machine learning algorithms and morphological analysis were the main techniques used in the early stages of skin lesion identification. To help dermatologists diagnose skin conditions, Moldovanu et al. [11] offered a skin lesion classification approach that integrates the dimensionality of surface fractals and statistics data on color clustering using two classifiers, k nearest neighbors and neural networks. Chatterjee et al. [12] suggested classifying skin lesion kinds using fractal-based feature extraction, morphological preprocessing, and recursive feature removal to facilitate the more precise diagnosis and treatment of skin lesions in the medical field. Ranjan et al. [13] introduced a method using a machine learning model to measure and score the severity of radiation dermatitis, while also analyzing its mathematical characteristics, including erythema’s shape, size, and color. As computer vision technology continues to advance, an increasing number of deep learning methods are being utilized in the recognition of medical diseases. Pan et al. [25] proposed an Ensemble-3DCNN model to analyze brain magnetic resonance imaging (MRI) to confirm the regularity of the pathological progression of Alzheimer’s disease. This model provided robust support for early diagnosis, disease monitoring, and the development of treatment strategies. Javadi Moghaddam et al. [26] introduced an adapted version of the DenseNet-121 COVID-19 diagnostic framework to achieve the precise identification of COVID-19 from X-ray images. This study introduced a novel technical tool for clinical diagnosis. Noorbakhsh et al. [27] employed CNNs for cross-category analysis to uncover conserved spatial behavior in tumor histology images, involving the processing and analysis of various types of tumor histology images using deep learning methods to reveal their similarities and differences.

Many deep learning models have been created and shown to be effective in the field of identifying diseases early on. Thieme et al. [28] introduced a deep convolutional neural network (named MPXV-CNN) for identifying characteristic skin lesions caused by the monkeypox virus, achieving a sensitivity of 89% in the prospective cohort. Anand et al. [16] introduced a fusion model that integrates the U-Net and CNN models to accurately identify skin lesions. The model was simulated and analyzed using the HAM10000 dataset, which includes 10,015 dermoscopic images, achieving an accuracy of 97.96%. Gomathi et al. [29] introduced a novel dual optimization approach based on a deep learning network for skin cancer detection. The proposed network achieved an accuracy of 98.76% when evaluated with the HAM10000 dataset. Due to the inadequate availability of training datasets for disease recognition in real-world application scenarios, transfer learning is often utilized to address this issue. Mahbod et al. [30] proposed and evaluated a multi-scale multi-CNN fusion approach for skin lesion classification based on pre-trained CNNs and transfer learning, resulting in an accuracy of 86.2% on the ISIC 2018 dataset. Karri et al. [31] conducted a study on a two-phase cross-domain transfer learning approach, which involved both model-level and data-level transfer learning. They fine-tuned the system using two datasets, MoleMap and ImageNet, and achieved a Dice Similarity Coefficient (DSC) of 94.63% and 99.12% accuracy on the HAM10000 dataset. The results demonstrated the effectiveness of their approach in transferring knowledge across different domains. Additionally, attention was paid to image augmentation techniques to augment the diversity of the dataset, thereby enhancing recognition precision. Eduardo et al. [32] introduced a robust progressive growth of adversarial networks based on residual learning and recommended for facilitating the training of deep networks. This architecture could generate realistic synthetic 512 × 512 skin images, even when using small dermoscopic and non-dermoscopic skin image datasets as problem domains. With the rapid evolution of vision transformer technology, there has been a notable increase in its utilization for medical image recognition in recent years. This adaptation represents a significant leap forward in the integration of advanced transformer models to enhance the analysis and understanding of medical imaging data. Xin et al. [21] presented a skin cancer identification method based on a multi-scale vision transformer approach, which they further optimized using comparative learning. This technique achieved an impressive 94.3% accuracy on the HAM10000 dataset, demonstrating its effectiveness in the field. He et al. [33] introduced a Fully Transformer Network (FTN) to extract long-range contextual information for skin lesion analysis and successfully validated its effectiveness and efficiency on the ISIC 2018 dataset. Zhang et al. [24] developed a series of dual-branch hierarchical multi-modal transformer (HMT) blocks to systematically integrate data from multiple imaging modalities. This approach resulted in a diagnostic accuracy of 80.03% and an average accuracy of 77.99% on the Derm7pt dataset. In the realm of multi-modal skin disease identification, numerous studies have employed deep learning and vision transformer methods to fuse characteristics derived from skin images and patient-related metadata. Omeroglu et al. [23] proposed a multi-modal deep learning framework for skin lesion classification that integrated features of clinical images, dermoscopic images, and patient metadata. The framework employed three branches to extract features in a hybrid manner, achieving an average accuracy of 83.04% on the even-point criteria evaluation dataset, representing a 2.14% improvement over existing methods. He et al. [20] designed a cross-attention (CA) module that enabled collaboration between two modalities of dermoscopic images and clinical images through a cross-modal attention mechanism, enhancing the representation ability of features. On the seven-point criteria evaluation dataset, the average accuracy achieved was 76.8%, which was superior to the state-of-the-art methods.

This paper seeks to improve the recognition of multi-modal skin images by examining the effects of diverse deep learning networks. This study also delves into the use of transfer learning methods to enhance feature extraction from images with varying modalities. Furthermore, to expand the training dataset, a variety of data augmentation techniques are employed. This comprehensive approach aims to advance the comprehension and utilization of deep learning in multi-modal skin image analysis.

## 3. Methods

Figure 2 illustrates the holistic architecture of the MDSIS-Net (Figure 2a) network proposed in this study. The network comprises several key modules, including the image preprocessing module (Figure 2a), the multi-modal intra-feature extraction module (Figure 2a), the multi-modal information-switching module (ISM) (Figure 2b), and the feature aggregation module (Figure 2a). VISIA imaging is used to collect five modal images of patients with DD as input images. These modal images include spots, red marks, UV spots, porphyrins, and brown spots. MM images consist of two modalities of images: clinical images and dermoscopic images. The image preprocessing module is essential for refining input data through normalization and augmentation to guarantee optimal data quality and compatibility with the network. This study focuses on extracting meaningful features from different modalities in the input data through a multi-modal intra-feature extraction module. By leveraging multi-modal information, this module effectively captures diverse and complementary data, enhancing the network’s discriminative capabilities. To achieve this, we utilize the EfficientNetV2 block as the fundamental network module for multi-modal intra-feature extraction [34]. It consists of eight blocks, each producing feature maps at different scales, and employs shared parameters for handling multi-modal inputs. To enhance integration and information exchange across different modalities, a multi-modal ISM is integrated. This module strategically chooses and merges the most informative features from each modality, enabling effective information fusion and maximizing the synergistic advantages provided by multiple modalities. Additionally, it serves as an interactive platform for different modality features, enabling effective collaboration and exchange of information. The ISM further facilitates the exchange of information among the eight different scales of feature maps extracted by the EfficientNetV2. It ensures seamless communication and interaction between the feature maps at various scales, promoting effective information fusion and integration. Finally, the feature aggregation module consolidates the extracted features from different modalities into a unified representation. This comprehensive architecture enables the network to tackle the complexities of the given task and enhance its predictive capabilities. The model will be validated using the DD and MM datasets.

Algorithm 1 presents an illustration of the workflow for our proposed MDSIS-Net model.
**Algorithm 1** The pipeline for our proposed MDSIS-Net modelInput: Multi-modal image datasets of MM and DDOutput: Predicted classes and Grad-CAMSplitting the dataset: Every dataset is separated into testing, validation, and training sets.1. Training phase: Hyperparameter:   Image size: (384, 384, 3)   Number of input modalities: 5 and 2   Size of training batch: 32   Initial learning rate: 0.05   Decay type: CosineAnnealingLR   Optimizer selection: SGD Training runtime:   for epoch in range(begin_epoch, end_epoch)    Multi-modal image normalization    Multi-modal image augmentation    Multi-modal intra-feature extraction    Multi-modal information switching    Feature aggregation and classification    Predict the category and obtain Grad-CAM    Calculate the weighed cross-entropy loss    Gradient update and backpropagation2. Testing Read the image pixel Data normalization Feed the best training model Obtain the predicted mask and feature map Computer the metric: mAP, accuracy, precision, recall, and f1-score3. Inference runtime Obtain an image Data normalization Feed the best model Obtain the predicted class and feature maps

### 3.1. Multi-Modal Skin Lesion Dataset

Two datasets are used to validate the effectiveness of our proposed MDSIS-Net model, one derived from the publicly available dataset Derm7pt for multi-modal MM recognition, and the other derived from clinical VISIA images for DD identification. A comprehensive dataset has been collected, which includes 2005 samples totaling 10,025 VISIA images with five distinct modalities: spots, red marks, UV spots, porphyrins, and brown spots. This study was approved by the Ethics Committee of the First Affiliated Hospital of Ningbo University on 20 December 2023 (approval No. 2023R-178RS). All participants were informed about the right to withdraw from the study at any time, following the ethical standards of the Declaration of Helsinki, revised in 2013. The spot modality utilizes standard white light imaging to capture visible skin surface pigmentation. Red marks can reflect the condition of capillaries. UV spots are captured using 365 nm ultraviolet light, reflecting potential pigmentation beneath the epidermis, which correlates positively with skin photoaging. Porphyrins exhibit fluorescence, particularly in the T-zone, and are metabolic byproducts of bacteria residing in the follicular openings. Brown spots represent deeper, more latent pigmentation than UV spots. These images have been categorized into dermatitis, melasma, and acne. Figure 3 demonstrates a representative example of the VISIA multi-modal dataset. The clinical labels for the VISIA dataset are provided by experienced dermatologists, who assign diagnostic labels based on the lesion responses observed across different modalities. The Derm7pt dataset is used for the study of the diagnosis and classification of MM and NMSC. It consists of both clinical and dermoscopic images. Clinical images capture the macroscopic lesion characteristics of MM patients, while dermoscopic images provide a macroscopic view of the lesion features in MM patients. This dataset contains 1011 samples, totaling 2022 two-modal images, with 252 cases of MM and 759 cases of NMSC. The images in this dataset are annotated according to a seven-point checklist criteria. Figure 4 demonstrates a representative example of the Derm7pt multi-modal dataset. Both datasets have been randomly split into training, validation, and test sets, with the respective allocations being 70%, 15%, and 15%, to facilitate model development and evaluation. Subsequently, the training set will be employed for deep learning model training, the validation set for fine-tuning hyperparameters, and the test set for rigorously validating the model’s accuracy and generalization. Table 1 lists the detailed data distribution and purpose for the DD and MM datasets.

### 3.2. Multi-Modal Image Preprocessing

In multi-modal image recognition, augmentation and normalization play crucial roles. Normalization is essential for standardizing the image data from various modalities to a specific range of values, ensuring consistency in the data representation. Additionally, augmentation is valuable for increasing the diversity of the training data, thereby enhancing the model’s ability to generalize to a wide range of input variations.

#### 3.2.1. Multi-Modal Image Normalization

The dermatological dataset studied in this paper consists of multiple modalities, each with distinct pixel value ranges. Therefore, normalizing each modality is crucial to standardize the data and ensure fair comparison across modalities. This normalization process is essential to facilitate subsequent feature extraction and guarantee that the significance of each modality is appropriately considered, rather than favoring the modality with the larger pixel values. The normalization will allow for a more accurate and comprehensive analysis of the dataset, leading to better insights and potential improvements in skin disease diagnosis and treatment. There are numerous deep learning-based normalization methods utilized for image recognition, each offering distinct benefits and applications. Robust scaling, Z-score normalization, image-specific normalization, and min–max scaling are a few popular normalization methods. Z-score normalization is particularly beneficial in standardizing the scales of features, reducing the effects of varying ranges across features and improving the model’s capacity for learning and precise forecasting. It also helps in mitigating the impact of outliers and extreme values, making it a robust normalization technique suitable for a wide range of image recognition tasks. This technique calculates the Z-score (mean minus the standard deviation) of each channel and scales the values to a common range, such as [0, 1]. It serves as a robust method for normalization as it minimizes the impact of varying pixel values across different skin conditions. In this study, we utilize the following formula for the z-score normalization of each modality.(1)$$Z_{i}=\frac{X_{i}-\mu_{i}}{\sigma_{i}}$$

$X_{i}$ represents the raw value of each mode, which includes the channels for red, green, and blue, where i represents the index of all modes. $X_{i}$ is a vector of $\left[R_{i},G_{i},B_{i}\right]$. $\mu_{i}$ represents the average of the raw data and is a vector of $\left[\mu_{r_{i}},\mu_{g_{i}},\mu_{b_{i}}\right]$. $\sigma_{i}$ indicates the input data’s standard deviation and is a vector of $\left[\sigma_{r_{i}},\sigma_{g_{i}},\sigma_{b_{i}}\right]$. $Z_{i}$ represents the value after normalization and is a $\left[Z_{r_{i}},Z_{g_{i}},Z_{b_{i}}\right]$ vector. The formula representing $\mu_{i}$ is as follows.(2)$$\begin{array}{l} \mu_{r_{i}}=\frac{1}{N}{\sum R}_{i}\\ \mu_{g_{i}}=\frac{1}{N}\sum G_{i}\\ \mu_{b_{i}}=\frac{1}{N}\sum B_{i}\; \end{array}$$

Here is the formula for $\sigma_{i}$.(3)$$\begin{array}{l} \sigma_{r_{i}}=\sqrt{\frac{1}{N}{\sum \left(R_{i}-\mu_{r_{i}}\right)}^{2}}\\ \sigma_{g_{i}}=\sqrt{\frac{1}{N}{\sum \left(G_{i}-\mu_{g_{i}}\right)}^{2}}\\ \sigma_{b_{i}}=\sqrt{\frac{1}{N}{\sum \left(B_{i}-\mu_{b_{i}}\right)}^{2}} \end{array}$$

#### 3.2.2. Multi-Modal Image Augmentation

Image augmentation is crucial for deep learning training and the recognition of skin lesions in multi-modal images. This technique improves image quality, making it more suitable by enhancing visual effects. Skin lesion features may be affected by factors such as lighting, color, and contrast, necessitating image augmentation to clarify and highlight these features. By increasing contrast, reducing noise, and adjusting color balance, image augmentation improves the clarity and distinction of skin disease features, aiding deep learning algorithms in better recognition and understanding them. Moreover, image augmentation ensures image consistency, facilitating comparison and matching between images of different modalities, thereby improving the generalization ability and recognition accuracy of deep learning algorithms. It also mitigates class imbalance and enhances data diversity. Table 2 enumerates the image augmentation techniques applied in this paper and offers a thorough explanation of the associated parameters.

### 3.3. Multi-Modal Intra-Feature Extraction

This study utilizes EfficientNetV2, an improved multi-modal feature extraction network that is based on EfficientNet. By adjusting parameters such as depth, width, and input image resolution, EfficientNetV2 enhances network performance. It integrates neural architecture search and a composite model expansion method to develop a classification recognition network. Through the selection of optimal composite coefficients, the network’s depth, width, and input image resolution are proportionally expanded in three dimensions. This search for optimal coefficients aims to maximize recognition feature accuracy. By dynamically balancing these three dimensions, the number of coefficients and complexity during model training is effectively reduced, leading to a significant improvement in model performance. This approach yields better results compared to single-dimension scaling and also contributes to faster training speeds. The network primarily consists of 3 × 3 convolution, stacked Fused-MBConv, MBConv, and 1 × 1 convolution modules. The network structure details are provided in Appendix A, which consists of eight stages that generate feature maps at distinct scales. These eight feature maps at various scales serve for the subsequent ISM to extract intricate relationships between different modalities. In this research, transfer learning is employed for initialization on the ImageNet1K dataset to obtain preliminary parameters, and shared parameters are used to handle multi-modal inputs, resulting in a coherent and concise representation.

The MBConv module consists of a 1 × 1 ascending convolution, a 3 × 3 depth-wise convolution, a Squeeze-and-Excitation (SE) attention module, and a 1 × 1 descending convolution, as depicted in Appendix A. On the other hand, the Fused-MBConv module replaces the 1 × 1 ascending convolution and 3 × 3 deep convolution in MBConv with a simpler 3 × 3 convolution structure, as illustrated in Appendix A.

Table 3 presents the operation type, kernel size, stride size, expand ratio, output channel size, and layer size for each block. The expand ratio represents the amplification factor for the middle channel in each module.

### 3.4. Multi-Modal Information-Switching Module

When conducting deep learning-based skin lesion recognition, the incorporation of different modality images provides a more comprehensive and diverse set of information, thereby aiding in the precise identification of various types of skin diseases. The five VISIA modality images consist of spots, red marks, UV spots, porphyrins, and brown spots. Spot images depict small spots on the skin, red mark images illustrate areas with red patches, UV spot images capture spots induced by UV rays, porphyrin images highlight the presence of porphyrins in the skin, and brown spot images showcase brown patchy areas on the skin. In the case of MM, dermoscopic images allow for the visualization of its microscopic lesions, such as those in the vascular region, and clinical images allow for the observation of the features of MM at the macroscopic level, such as contours and overall color. The two modalities complement each other. By harnessing the power of multi-modal information exchange, we effectively communicate and integrate diverse features to accurately classify and differentiate among different skin diseases. Appendix A demonstrates the specific framework of ISM; each modality goes through a feature extraction network to obtain eight stages of feature maps, and each stage of feature maps goes through the ISM module for information exchange, where i denotes the index of different modalities, n indicates the number of all modalities, and s represents the stage of feature extraction.

Given an input image I, consisting of *n* modalities $I=\{I_{1},I_{2},\ldots ,I_{n}\}$, we begin by feeding the image of each modal a basic network block $F_{b}$ to generate multi-modal features $X_{i}^{s}$, where s represents the stage index of feature extraction. The formula for multi-modal feature extraction is expressed as follows.(4)$$X_{i}^{s}=F_{b}(I_{i}),X_{i}^{s}\in R^{c_{s}\times h_{s}\times w_{s}}$$

Next, the feature maps $X_{i}^{s}$ are fed into the ISM for feature exchange. ISM employs Multi-head Self-Attention (MSA) across features of different modalities. For the features at each spatial location, we consider the features from different modalities as a sequence that requires information exchange. Specifically, for the feature maps $X_{i}^{s}$, they are stacked together and transformed into $X^{s}$, meeting the input format required by MSA using Permute and Reshape operations. The ISM initially employs three separate fully connected (FC) layers to map it to the queries (Q), keys (K), and values (V) for the *j*-th attention head as follows.(5)Xs=Reshape(Permute(Stack(X1s,X2s,…,Xns))),Xs∈R(hs×ws)×n×cs(6)$$Q_{j}^{s}=FC1_{j}^{s}(X^{s})$$(7)$$K_{j}^{s}=FC2_{j}^{s}(X^{s})$$(8)$$V_{j}^{s}=FC3_{j}^{s}(X^{s})$$

Subsequently, we compute the attention for each head of every modality as follows.(9)$$H_{j}^{s}=Attention(Q_{j}^{s},K_{j}^{s},V_{j}^{s})=soft\max(\frac{Q_{j}^{s}{K_{j}^{s}}^{T}}{\sqrt{d}})V_{j}^{s}$$

d represents the dimension of each head. Through the concatenation of the attention outputs from m heads and the utilization of the FC layer, the output of the MSA is obtained, which has the same dimension as the input sequence. This method then utilizes Permute and Reshape operations, as well as Spit operations, to obtain the information exchange features $Y_{i}^{s}\in R^{c_{s}\times h_{s}\times w_{s}}$ under different modalities. The formula is as follows.(10)$$H^{s}=Concat(H_{1}^{s},H_{2}^{s},\ldots ,H_{m}^{s}),H^{s}\in R^{\left(h_{s}\times w_{s})\times n\times (m\times c_{h}\right)}$$(11)$$Y^{s}=FC(H^{s}),Y^{s}\in R^{(h_{s}\times w_{s})\times n\times c_{s}}$$

Finally, the input multi-modal feature maps and the information-exchanged multi-modal feature maps are linearly summed together using a linear summation approach, and the summation result serves as the output $Z_{i}^{s}$ of the ISM module for subsequent feature aggregation and classification. The formula is as follows.(12)$$Z_{i}^{s}=F_{s}(X_{i}^{s},Y_{i}^{s})=\alpha X_{i}^{s}+(1-\alpha )\sum_{j\neq i}Y_{j}^{s}$$

The skin disease feature images of different modalities undergo a comprehensive process of information exchange, encompassing eight stages. This progressive information exchange facilitates the extraction of intricate feature relationships between the diverse modalities to a greater extent. The outcome of this process is a final feature map, which serves as a vital resource for the subsequent steps of feature fusion and classification.

### 3.5. Feature Aggregation and Classification

In this study, a merging technique is employed to fuse the feature maps obtained from the information exchange of different modalities. This fused feature map is then utilized for the subsequent classification of skin lesions. Appendix A illustrates the specific process of feature fusion and classification. A 1 × 1 convolutional layer is employed to reduce the dimensionality of the feature maps obtained from the information exchange. Simultaneously, global average pooling (GAP) is used to compute the average of the two-dimensional feature maps along each channel. Consequently, one-dimensional vectors are derived. The feature vectors extracted from multiple modalities are fused using concatenation. Subsequently, the fused vector is processed by an FC layer and softmax activation to obtain the predicted probabilities for skin lesion classification.

As shown in Equations (13) and (14), each modality feature map $Z_{i}$ obtained from the information exchange is individually processed through GAP and a 1 × 1 convolutional layer. The resulting feature maps are then concatenated across all modalities to obtain a one-dimensional feature vector M. This feature vector is then fed into an FC layer and undergoes softmax activation to generate the predicted probability P for skin disease classification.(13)$$M=cat(GAP(Conv(Z_{1})),\ldots ,GAP(Conv(Z_{n})))$$(14)P=softmax(FC(M))

During the training phase of this research, Weighted Cross Entropy Loss (WCEL) is employed as the loss function for the multi-modal skin disease classification model. WCEL is a technique used to address class imbalance or the varying importance of different classes. The mathematical formula for WCEL is as follows.(15)$$L=-\sum w_{k}y_{k}\log P_{k}$$

The weight assigned to class k is denoted by $w_{k}$, while $y_{k}$ represents the true label for class k (0 or 1) and $P_{k}$ signifies the predicted probability of class k. The weights $w_{k}$ can be determined based on the class distribution in the dataset or manually defined to emphasize or de-emphasize specific classes. Loss is calculated for each class and then aggregated to obtain the overall loss value, which, in turn, is utilized to optimize the model during the training process.

### 3.6. Experimental Settings

In this research, a deep learning platform for multi-modal skin lesion identification is constructed using PyTorch V2.0.1. The experimental setup consists of two GPUs with a memory capacity of 12 GB each and an Intel(R) i7 CPU. The network is trained using images resized to 384 × 384 pixels, and the Stochastic Gradient Descent (SGD) optimizer is employed. The initial learning rate is set to 0.05, and the training is performed with a batch size of 32. This hardware and software configuration enables the efficient processing and optimization of the model for accurate skin lesion recognition. The detailed model configuration is shown in Table 4.

#### Evaluation Metrics and Results

Five performance criteria are used in this study to assess our multi-modal skin disease recognition model: f1-score, mean Average Precision (mAP), recall, accuracy, and precision. These measurements offer a thorough grasp of the model’s performance in correctly categorizing skin conditions. Precision indicates the proportion of correctly classified positive samples out of all samples predicted as positive. It is calculated as the ratio of true positives (TP) to the sum of true positives and false positives (FP). The formula for precision is as follows.(16)$$Precision=\frac{TP}{TP+FP}$$

Recall gauges the model’s capacity to accurately distinguish positive samples from all of the real positive samples. It is sometimes referred to as sensitivity or true positive rate. The ratio of true positives (TP) to the total of false negatives (FN) is used to compute it. The following is the recall formula.(17)$$Recall=\frac{TP}{TP+FN}$$

By computing the ratio of correctly classified samples to all samples, accuracy evaluates how accurate the model’s predictions are overall. TN refers to the true negatives, which are the number of samples that are correctly predicted as negatives. It is calculated using the following formula.(18)$$Accuracy=\frac{TP+TN}{TP+TN+FP+FN}$$

F1-score is a balanced evaluation metric that takes both precision and memory into account. It is calculated as the harmonic mean of precision and recall. It is computed with the following formula.(19)$$F1\mathrm{-s}core=\frac{2*Precision*Recall}{Precision+Recall}$$

The mAP evaluates the model’s performance in multi-class classification problems. It computes the Average Precision for each class and then takes the mean across all classes. It serves as a robust measurement of the model’s classification accuracy across multiple classes. The formula for mAP is as follows.(20)$$mAP=\frac{AP_{1}+AP_{2}+\ldots +AP_{n}}{n}$$

$AP_{i}$ represents the Average Precision for the class i. n is the total number of classes. These evaluation indicators of precision, recall, accuracy, f1-score, and mAP collectively provide insights into the model’s performance in terms of prediction correctness, sensitivity, overall accuracy, and classification accuracy across multiple classes.

## 4. Results and Assessment of the Experiments

### 4.1. Experimental Results for the Multi-Modal Skin Lesion Datasets

To evaluate the effectiveness of our multi-modal model, we compare it with the latest image classification model on the DD dataset using five performance metrics: mAP, accuracy, precision, recall, and f1-score. The results, as shown in Table 5, showcase the superior performance of our proposed MDSIS-Net. It achieves an mAP of 0.967, accuracy of 0.960, precision of 0.935, recall of 0.960, and f1-score of 0.947, outperforming all other models. MDSIS-Net yields significant improvements in mAP, with respective increases of 5.5%, 4.0%, 0.9%, 4.3%, 1.6%, and 1.5% compared to other methods. In terms of accuracy, MDSIS-Net exhibits improvements of 11.0%, 7.7%, 1.0%, 6.0%, 2.0%, and 4.3% over other methods. Additionally, MDSIS-Net outperforms other methods in precision, showing improvements of 10.5%, 6.8%, 0.9%, 5.5%, 1.3%, and 3.2%. Regarding recall, MDSIS-Net shows improvements of 8.3%, 6.7, 0.7%, 5.3%, 1.3%, and 3.3% over other models. Lastly, MDSIS-Net achieves the highest f1-score, surpassing other methods with improvements of 9.4%, 6.7%, 0.8%, 5.4%, 1.3%, and 3.3%. These results collectively demonstrate that MDSIS-Net consistently outperforms other models across all five metrics, providing strong evidence for its effectiveness in multi-modal image recognition. Furthermore, we provide a detailed comparison with the state-of-the-art multi-modal advanced transformer-based approach [35], and our model outperforms it in terms of performance on the DD dataset. The statistical significance of our model’s improvement on the DD dataset, validated by a paired *t*-test, is shown in Table 5 (*p*-values).

The test accuracy of several models on the DD dataset is shown in Figure 5. The orange curve corresponds to our proposed MDSIS-Net, which consistently outperforms other methods by achieving the highest convergence accuracy. This demonstrates the superior performance and efficacy of our MDSIS-Net model in accurately classifying DD in the dataset.

Likewise, we test the performance of our proposed model on the MM multi-modal dataset and compare it with some recent image classification models. The results, as shown in Table 6, clearly demonstrate the superior performance of our proposed MDSIS-Net. It achieves an mAP of 0.877, accuracy of 0.907, precision of 0.911, recall of 0.815, and f1-score of 0.851, outperforming all other models. MDSIS-Net consistently outperforms other methods by 7.7%, 2.7%, 11.2%, 8.6%, 4.8%, and 4.1% in terms of mAP. In accuracy, MDSIS-Net exhibits improvements of 6.6%, 3.3%, 7.3%, 5.3%, 3.9%, and 3.3% over other methods. Moreover, MDSIS-Net excels in precision, showing enhancements of 12.4%, 6.7%, 10.7%, 6.6%, 8.9%, and 4.4% compared to other models. The model also demonstrates superior performance in recall, with enhancements of 9.5%, 4.2%, 14.1%, 10.7%, 3.6%, and 6.3% over other models. Lastly, MDSIS-Net achieves the highest f1-score, surpassing other methods with improvements of 10.7%, 5.1%, 14.6%, 10.7%, 5.4%, and 1.5%. Overall, these findings highlight the consistent superiority of MDSIS-Net across all five metrics, providing compelling evidence for its effectiveness in multi-modal image recognition. Additionally, we conduct a comprehensive comparison with the state-of-the-art multi-modal advanced transformer-based approach [35], demonstrating that our model achieves superior performance on the MM dataset. The improvement of our model on the MM dataset, confirmed by a paired *t*-test, is statistically significant, as indicated in Table 6 (*p*-values).

The test accuracy of several models on the MM dataset is shown in Figure 6. The orange curve represents our proposed MDSIS-Net, which exhibits the highest convergence accuracy and consistently outperforms other methods. These results highlight the superior performance and effectiveness of our MDSIS-Net model in accurately classifying MM within the dataset.

### 4.2. Interpretability Analysis of Multi-Modal Skin Lesions

To analyze the interpretability of our proposed multi-modal recognition model, we utilize Gradient-weighted Class Activation Mapping (Grad-CAM) for visual analysis. Grad-CAM calculates the gradient information of the target class for the model’s output, multiplies the gradient with the feature map to obtain weights, and, finally, adds the weights to the feature map to generate the class activation map. This visualization technique allows us to identify the image regions that the model concentrates on during the decision-making process, enabling the human understanding of the model’s decision rationale. Figure 7 displays the Grad-CAM results of dermatitis. In Figure 7, the second row illustrates the Grad-CAM-generated heatmap, with lesion edges (shown by the blue boundaries) detected using the findContours algorithm based on heatmap intensity [40]. The third row shows the ground truth lesion regions of interest (ROIs) (shown by the red bounding boxes) annotated by dermatologists. In the VISIA “red marks” pattern, subtle skin barrier damage can be observed that may not be apparent to the naked eye, aiding in the assessment of the severity of dermatitis. The Grad-CAM visualization reveals an overlap between the prominent red area and the damaged skin barrier area. In some cases, such as the “red marks” on lips, the heatmap may exhibit misidentifications due to the similarity in color and texture between healthy lip regions and dermatitis-affected areas. This issue may arise because our lesion area segmentation is based on an unsupervised method. In the VISIA “UV spots” and “brown spots” patterns, pigment deposition areas are identified suggesting a potential risk of melasma. The fluorescent-prominent areas overlapped with the pigment deposition areas, confirming that the MDISB-Net model helps identify dermatitis patients, assessing their severity and skin lesion progression trends.

Figure 8 illustrates the Grad-CAM results of melasma, with the more prominent red and fluorescent areas indicating the severity of melasma. It is evident from the Grad-CAM images of the “red masks”, “UV spots”, and “brown spots” that the reflection of the melasma area is most pronounced. The appearance of some red spots in the Grad-CAM images indicates that patients with melasma also have an impaired skin barrier, while also showing that the pigment deposition area is larger than what is visible to the naked eye, signifying an advanced stage of melasma. The heatmap analysis aligns well with the diagnoses of dermatologists, further confirming the interpretability of the MDISB-Net model in identifying melasma. This interpretability enhances the trustworthiness and credibility of the MDISB-Net model in accurately identifying melasma.

Figure 9 showcases the Grad-CAM visualization of acne, where deeper shades of red indicate a more severe condition of acne. The coverage extends beyond the visible skin lesions present to include areas where acne and seborrheic dermatitis have occurred in the past. The heatmap analysis aligns closely with the clinical assessments of dermatologists, confirming the interpretability of the MDISB-Net model in detecting acne. This can assist dermatologists in establishing more accurate diagnoses, optimizing treatment strategies, and enhancing care for patients managing acne.

Figure 10 presents the Grad-CAM visualization of malignant melanoma (MM), with darker areas indicating the specific characteristics of MM. The clinical image Grad-CAM provides an overview of the overall characteristics of MM. In the dermoscopic image Grad-CAM, the darker areas reveal irregular pigment networks, while the redder areas highlight the presence of the blue-white curtain associated with MM. It is worth noting that an artificial object in Figure 10 is marked as a significant feature in the heatmap, which may be attributed to the similarity in visual patterns between the object and MM characteristics. Despite this, the heatmap analysis closely aligns with clinical assessments conducted by dermatologists. This can aid dermatologists in making more accurate diagnoses, optimizing treatment strategies, and improving care for patients managing MM.

Figure 11 displays the Grad-CAM visualization of NMSC, with darker areas indicating characteristics of clack nevus, a type of NMSC. The redder area in Grad-CAM highlights some plaque characteristics, offering valuable insights for clinical decision making regarding MM and NMSC.

### 4.3. Performance Analysis of Multi-Class Skin Lesion Recognition

To further investigate and visualize the classification results, this study utilizes t-distributed Stochastic Neighbor Embedding (t-SNE), a widely used nonlinear dimensionality reduction and visualization technique. The primary objective of t-SNE is to reduce the dimensionality of high-dimensional data while maintaining the relative distance relationships between samples. t-SNE is particularly effective in capturing local structures and clustering patterns in high-dimensional data. The DD dataset, with multiple modalities, is mapped into a lower-dimensional space using t-SNE, and a new probability distribution is constructed in this lower-dimensional space to resemble the distribution in the high-dimensional space as closely as possible. By visualizing the t-SNE plot in Appendix A, it can be observed that our multi-modal recognition model effectively distinguishes dermatitis, melasma, and acne, with acne being completely separated from the other two categories. There is a certain similarity between dermatitis and melasma, which could be attributed to their morphological similarities in clinical presentations. For example, both conditions involve skin lesions and can sometimes present with inflammation and pigmentation. The t-SNE results provide valuable insights into the classification performance of our model and the potential morphological similarities between dermatitis and melasma, reinforcing their association with the clinical characteristics of these disorders. Our proposed model has demonstrated efficiency in distinguishing the features of MM and NMSC, as depicted in the t-SNE plot in Appendix A, where MM features tend to cluster in the inner layer, while NMSC features are distributed around the outer layer of MM. There are some early cancer features present in MM that are more similar to the precancerous features in NMSC.

To evaluate the performance and accuracy of the MDSIS-Net model, we employ a confusion matrix (CM) to assess the model’s performance and accuracy, highlighting the correspondence between the actual category and the predicted category by the model. Appendix A displays the CM map of our model on the DD dataset. Acne can be distinguished from the other two categories relatively well, while there is some confusion between melasma and dermatitis. Appendix A displays the CM map of our model on the MM dataset. NMSC is largely correctly identified, with some misidentification of early cancer features in MM, which are more similar to precancerous features in NMSC.

## 5. Discussion

The current state of research in multi-modal skin lesion recognition has shown promising results, particularly with the integration of deep learning techniques [20,24,35]. Recent studies have highlighted the potential of combining various imaging modalities, such as dermoscopy, clinical images, and reflectance confocal microscopy, to improve the diagnostic accuracy of skin lesions [41]. However, challenges remain in effectively integrating these diverse modalities due to differences in data representation, resolution, and feature extraction requirements [42].

The proposed MDSIS-Net model addresses the challenges of integrating diverse imaging modalities for accurate multi-modal skin lesion recognition by employing a deep multi-modal information-switching network that facilitates the intricate interplay between these modes and enables the critical exchange of multi-scale feature information, thereby advancing the examination of skin ailments through end-to-end recognition. A novel multi-scale information-switching structure is developed within MDSIS-Net to facilitate the exchange of deep and shallow features between different modalities. This structure automatically adjusts the weight of information exchange at different feature layers, while also enhancing the inter-modality association of deep and shallow features within individual modalities. This optimizes the distribution of the distinctive features of different diseases across diverse imaging modalities. MDSIS-Net leverages deep convolutional neural networks to extract fine-grained features of skin lesions from various skin image modalities. Moreover, it enhances the information exchange between different modality features at various scales, thereby improving the interpretability and recognition accuracy of skin lesions in multi-modal skin imaging. The experiment is based on actual clinical data and public dataset Derm7pt. For DD, VISIA equipment is used to perform the multi-modal imaging of patients. Each patient’s images include five modalities: spots, red marks, UV spots, porphyrins, and brown spots. The melanoma dataset includes both clinical images and dermoscopic images. Our proposed model achieves an mAP of 0.967, accuracy of 0.960, precision of 0.935, recall of 0.960, and f1-score of 0.947 on the DD dataset, all surpassing the performance of the current best models. Furthermore, on the MM dataset, our proposed model achieves an mAP of 0.877, accuracy of 0.907, precision of 0.911, recall of 0.815, and f1-score of 0.851, all of which exceed the performance of the current best models. Furthermore, on the MM dataset, our proposed model achieves an mAP of 0.877, accuracy of 0.907, precision of 0.911, recall of 0.815, and f1-score of 0.851, all of which exceed the performance of the current best models. We conduct Grad-CAM for model interpretability analysis, and different modalities show hotspots that reflect the corresponding features of skin lesions in different modalities, which are similar to the focus areas of clinical diagnosis. Additionally, from the t-SNE and confusion matrix, we can see that our model has excellent discriminability and feature distinction abilities across multiple classes.

This study has several limitations. First, as the number of modalities increases, the multi-modal network’s computational complexity rises as well, requiring significant time and resources, which may limit its applicability in environments with limited resources. Initial tests using Automatic Mixed Precision (AMP) [43] training, which switches from float32 to float16 precision, show promise for optimization as they cut memory consumption in half with only a 1% accuracy loss. Second, Grad-CAM-based heatmap visualization, which performs unsupervised lesion region segmentation [44], may misidentify regions due to similarities in color, texture, or patterns between healthy and affected areas. Third, since image-based clinical diagnosis places high demands on image quality, the model’s performance is sensitive to factors such as blurring, noise, and size variations, which can affect results. While it supports DD and MM tasks, its generalizability to other skin diseases requires further validation.

## 6. Conclusions and Future Perspectives

In conclusion, this research introduces MDSIS-Net, a deep multi-modal information-switching network for the accurate diagnosis of skin lesions. To extract intra-modality features, MDSIS-Net uses a multi-scale fully shared convolutional neural network with transfer learning. It also includes a novel information-switching module that is improved by a cross-attention method. MDSIS-Net dynamically calibrates and integrates cross-modal features, enhancing inter-modality associations, feature representation, interpretability, and recognition precision. The model surpasses current methods and has been validated on both public datasets and actual clinical data. Dermatologists can greatly benefit from its interpretability, as evidenced by the Grad-CAM heatmaps that match clinical diagnostic focus areas. This work advances clinical decision making and lays a foundation for future developments in skin lesion diagnosis and treatment. In future work, we will focus on model compression and computation optimization to enhance the utilization of computing resources in multi-modal scenarios, enabling practical applications in clinical settings. Simultaneously, we will conduct in-depth research on patients with phenotypic skin lesions, delving into their living environments, habits, and pathogenic mechanisms to generate scientific discoveries, bridging the gap from phenotype to mechanism.

## Figures and Tables

**Figure 1 bioengineering-12-00282-f001:**
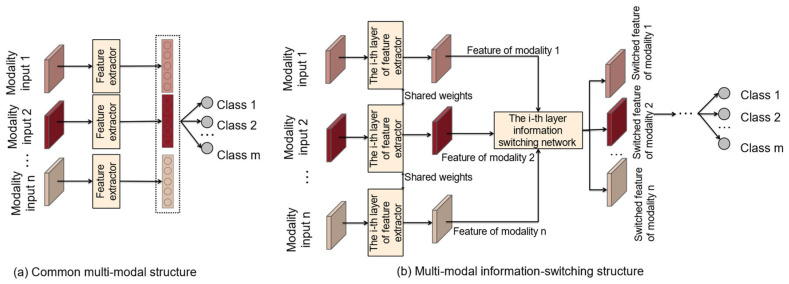
A demonstration of the differences between the deep multi-modal information-switching architecture and conventional multi-modal structures. Our method involves processing input images from different modalities through deep feature extraction and obtaining feature maps that are specific to each modality. Within each feature map layer, learnable parameters are applied to facilitate automatic information switching. Subsequently, the modified features undergo integration, and the consolidated features are used for the classification of disfiguring skin diseases.

**Figure 2 bioengineering-12-00282-f002:**
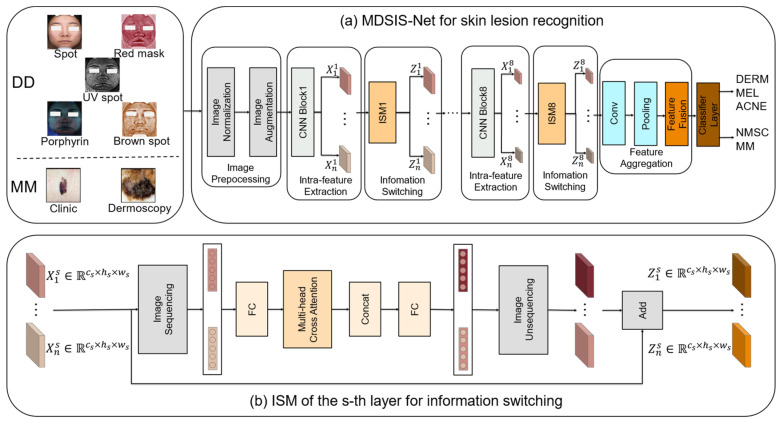
An overview of the MDSIS-Net model that we have proposed.

**Figure 3 bioengineering-12-00282-f003:**
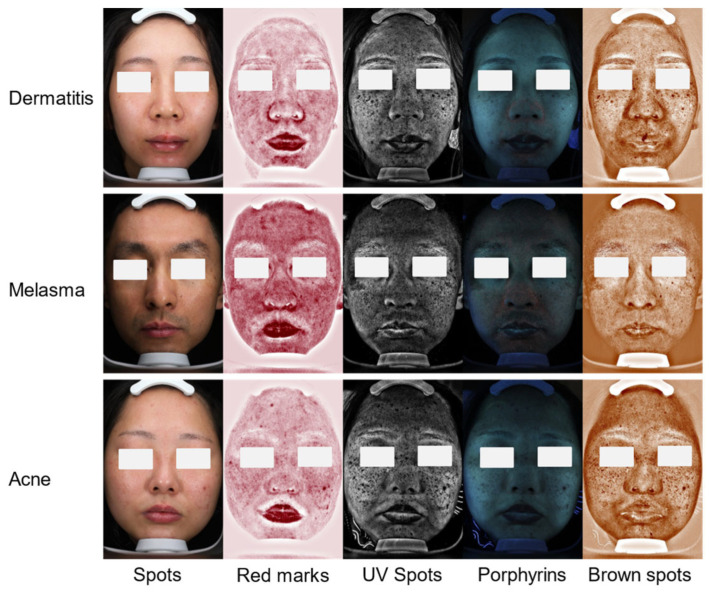
VISIA multi-modal representative images and their labels.

**Figure 4 bioengineering-12-00282-f004:**
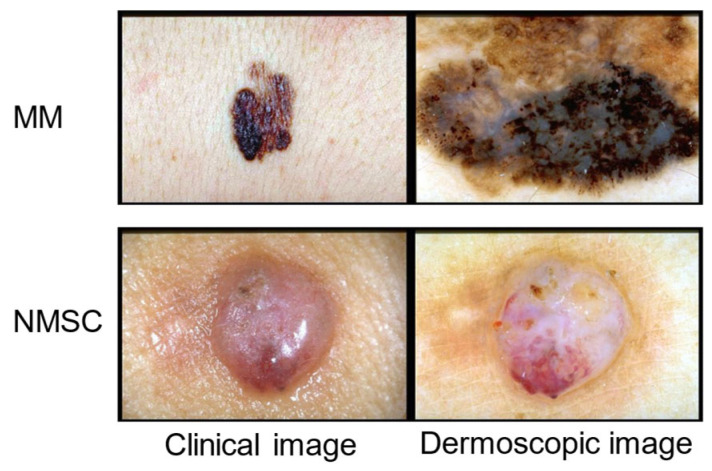
MM- and NMSC-representative images and their labels.

**Figure 5 bioengineering-12-00282-f005:**
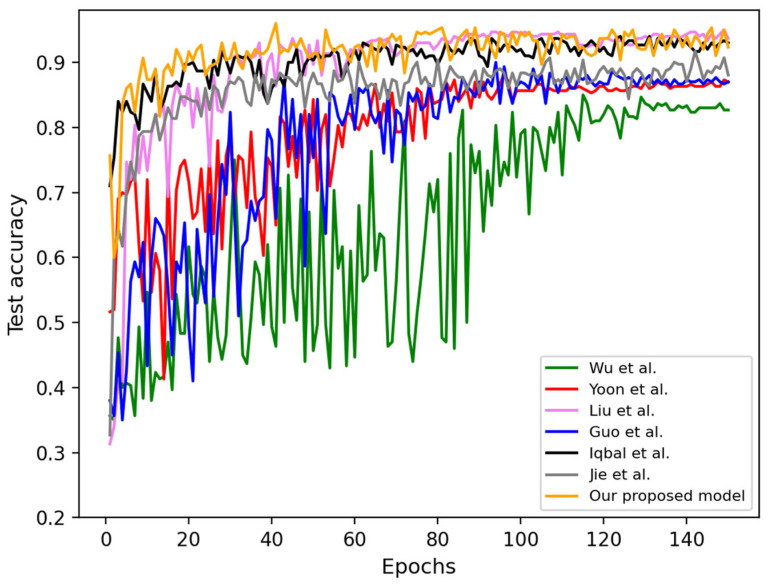
Seven models’ test accuracy curves on the DD dataset [9,35,36,37,38,39].

**Figure 6 bioengineering-12-00282-f006:**
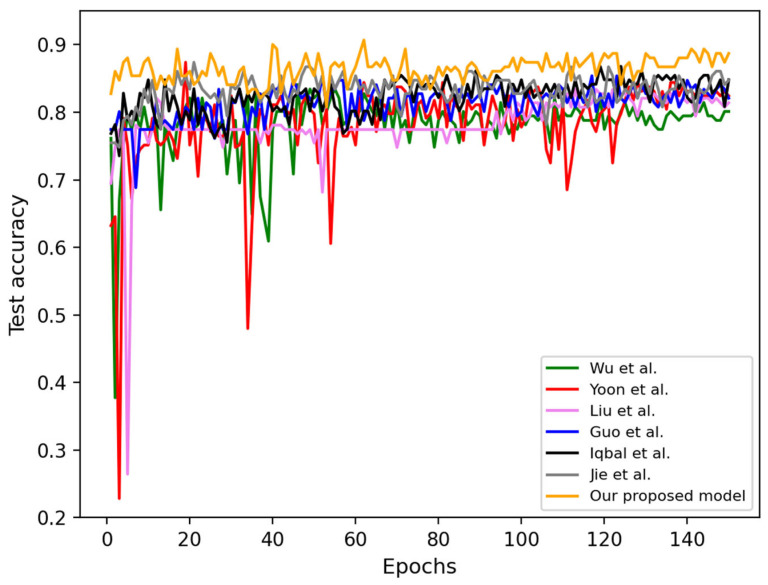
Seven models’ test accuracy curves on the MM dataset [9,35,36,37,38,39].

**Figure 7 bioengineering-12-00282-f007:**
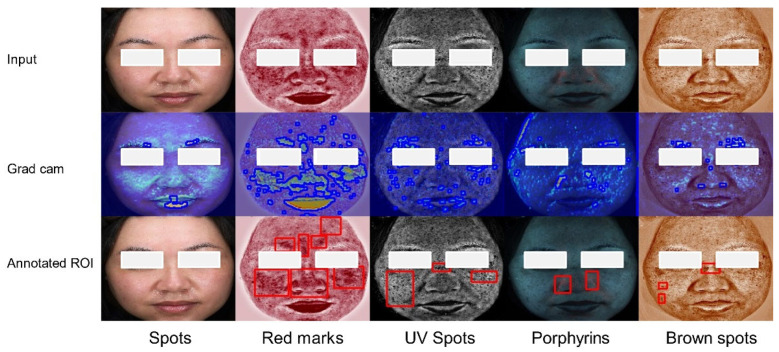
The feature map of the dermatitis class on the DD dataset across five modalities: original image (Row 1), Grad-CAM heatmap with lesion edges (blue boundaries) detected by findContours (Row 2), and ground truth lesion ROIs (red bounding boxes) annotated by dermatologists (Row 3).

**Figure 8 bioengineering-12-00282-f008:**
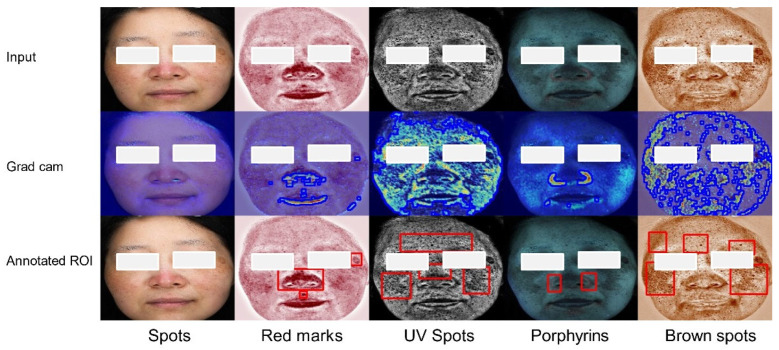
The feature map of the melasma class on the DD dataset across five modalities: original image (Row 1), Grad-CAM heatmap with lesion edges (blue boundaries) detected by findContours (Row 2), and ground truth lesion ROIs (red bounding boxes) annotated by dermatologists (Row 3).

**Figure 9 bioengineering-12-00282-f009:**
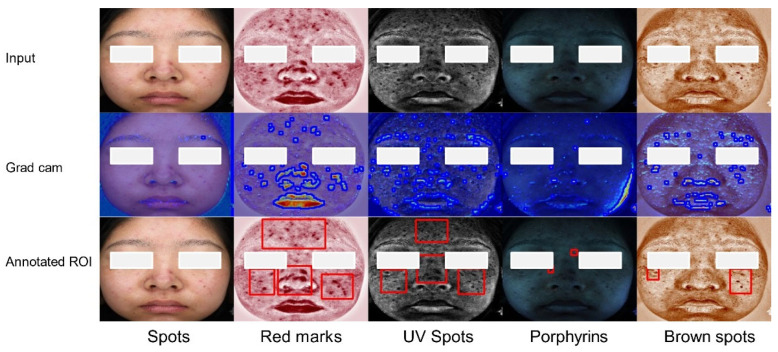
The feature map of the acne class on the DD dataset across five modalities: original image (Row 1), Grad-CAM heatmap with lesion edges (blue boundaries) detected by findContours (Row 2), and ground truth lesion ROIs (red bounding boxes) annotated by dermatologists (Row 3).

**Figure 10 bioengineering-12-00282-f010:**
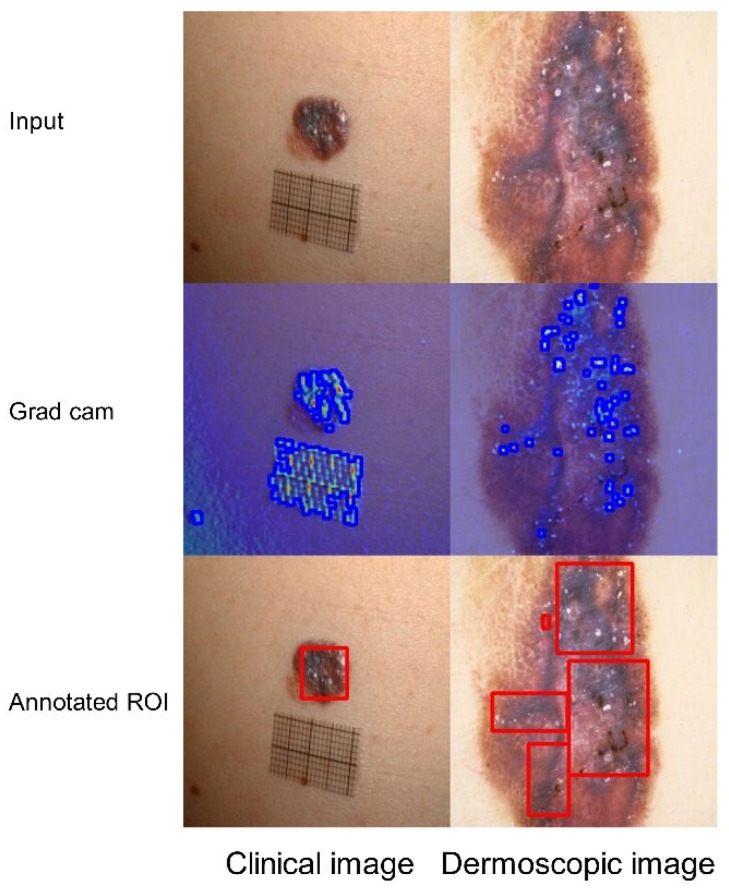
The feature map of the MM class on the Derm7pt dataset across two modalities: original image (Row 1), Grad-CAM heatmap with lesion edges (blue boundaries) detected by findContours (Row 2), and ground truth lesion ROIs (red bounding boxes) annotated by dermatologists (Row 3).

**Figure 11 bioengineering-12-00282-f011:**
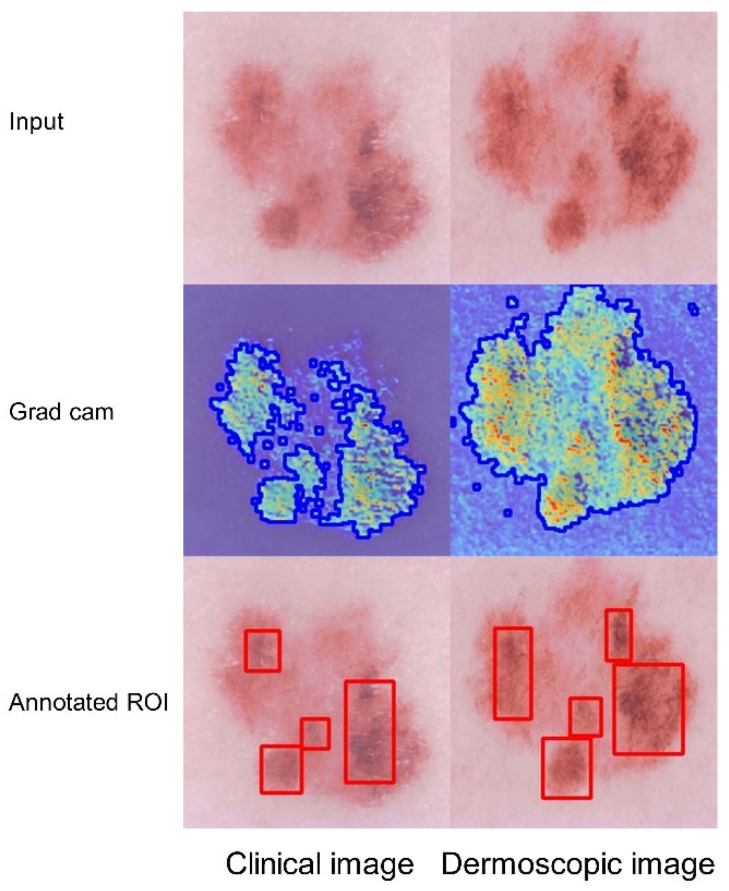
The feature map of the NMSC class on the Derm7pt dataset across two modalities: original image (Row 1), Grad-CAM heatmap with lesion edges (blue boundaries) detected by findContours (Row 2), and ground truth lesion ROIs (red bounding boxes) annotated by dermatologists (Row 3).

**Table 1 bioengineering-12-00282-t001:** Data distribution and purpose for the DD and MM datasets.

Dataset	Split	Number of Samples	Purpose
DD	Training	1403	Used for training our deep learning model on the DD dataset.
DD	Validation	302	Used to select the best-performing model, which is then evaluated on the DD test set.
DD	Test	300	Used to evaluate the performance of the model on the DD dataset.
MM	Training	707	Used for training our deep learning model on the MM dataset.
MM	Validation	153	Used to select the best-performing model, which is then evaluated on the MM test set.
MM	Test	151	Used to evaluate the performance of the model on the MM dataset.

**Table 2 bioengineering-12-00282-t002:** The comprehensive multi-modal image augmentation approaches utilized in this research.

Multi-Modal Image Augmentation	Parameters	Detailed Operations
Color jittering	From 0.8 to 1.2	With a chance of 0.2, the image’s brightness, contrast, and saturation are arbitrarily altered within the range of 0.8 to 1.2.
Random cropping	Probability of 0.3	The image is randomly cropped with a 30% probability while keeping the aspect ratio at 0.93.
Random flipping	Probability of 0.1	With a 10% probability, the image is symmetrically reflected in both vertical and horizontal directions at random.
Random rotating	From −20 to 20	The image is rotated at random, with a 20% chance of occurrence, between −20 and 20 degrees.

**Table 3 bioengineering-12-00282-t003:** The detailed parameters of the feature extraction network.

Stage	Operation	Kernel Size	Stride Size	Expand Ratio	Channel Size	Layer Size
1	Conv	3	2	1	32	1
2	Fused-MBConv	3	1	1	16	1
3	Fused-MBConv	3	2	4	32	2
4	Fused-MBConv	3	2	4	48	2
5	MBConv	3	2	4	96	3
6	MBConv	3	1	6	112	5
7	MBConv	3	2	6	192	8
8	Conv	1	1	1	1280	1

**Table 4 bioengineering-12-00282-t004:** Experimental configuration details of the MDSIS-Net model.

Category	Values/Configurations
Learning rate	0.05
Optimizer	SGD
Momentum	0.9
Batch size	32
Weight decay	0.0001
Epochs	150
Backbone	EfficientNetV2-B0
Total training time	2 h
Input size	384

**Table 5 bioengineering-12-00282-t005:** The classification results of the multi-modal DD dataset on the test set.

Model	mAP	Accuracy	Precision	Recall	F1-Score	*p*-Value (vs. Ours)
Wu et al. [36]	0.912	0.850	0.830	0.877	0.853	8.0 × 10^−4^
Yoon et al. [37]	0.927	0.883	0.867	0.893	0.880	5.2 × 10^−4^
Liu et al. [9]	0.958	0.950	0.926	0.953	0.939	7.2 × 10^−5^
Guo et al. [38]	0.924	0.900	0.880	0.907	0.893	4.4 × 10^−5^
Iqbal et al. [39]	0.951	0.940	0.922	0.947	0.934	4.0 × 10^−4^
Jie et al. [35]	0.952	0.917	0.903	0.927	0.914	2.3 × 10^−3^
Our proposed model	0.967	0.960	0.935	0.960	0.947	1.0

**Table 6 bioengineering-12-00282-t006:** The classification results of the multi-modal MM dataset on the test set.

Model	mAP	Accuracy	Precision	Recall	F1-Score	*p*-Value (vs. Ours)
Wu et al. [36]	0.770	0.841	0.787	0.720	0.744	4.9 × 10^−4^
Yoon et al. [37]	0.850	0.874	0.844	0.773	0.800	3.3 × 10^−3^
Liu et al. [9]	0.765	0.834	0.804	0.674	0.705	9.2 × 10^−4^
Guo et al. [38]	0.791	0.854	0.845	0.708	0.744	1.5 × 10^−3^
Iqbal et al. [39]	0.829	0.868	0.822	0.779	0.797	5.0 × 10^−3^
Jie et al. [35]	0.836	0.874	0.867	0.752	0.836	1.2 × 10^−3^
Our proposed model	0.877	0.907	0.911	0.815	0.851	1.0

## Data Availability

The raw VISIA multi-modal dataset supporting the conclusions of this study will be made available by the authors following the First Affiliated Hospital of Ningbo University’s institutional management and sharing policy. The Derm7pt dataset is publicly available at https://derm.cs.sfu.ca/Welcome.html (accessed on 6 March 2025).

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
