# Peer review of "Deep Multi-Modal Skin-Imaging-Based Information-Switching Network for Skin Lesion Recognition"

_bioengineering, 2025, doi:10.3390/bioengineering12030282_

Round 1

Reviewer 1 Report

Comments and Suggestions for Authors

This paper introduces the Multi-Modal Skin Imaging-based Information Switching Network (MDSIS-Net), a deep learning framework for skin lesion recognition. The framework leverages multi-modal imaging data and employs an information-switching module to facilitate feature exchange between different modalities and scales. The proposed model demonstrates superior performance on clinical dermatological datasets. This paper is easy to follow but may be limited by the following aspects.

0. The abstract is too long and is hard to follow. Besides, there is a significant typo error in Line 358 with the funny sentence "acne. Error! Reference source not found."

1. The paper lacks sufficient implementation details, such as specific hyperparameter values, model configurations, and training runtime, which may hinder reproducibility by other researchers.

2. Although the proposed model outperforms baselines, the paper does not provide a detailed discussion or comparison with other state-of-the-art multi-modal approaches [1] , especially those employing recently advanced transformer architectures.

[1] Jie, Y., Li, X., Tan, H., Zhou, F., & Wang, G. (2024). Multi-modal medical image fusion via multi-dictionary and truncated Huber filtering. Biomedical Signal Processing and Control88, 105671.

3. The paper does not discuss potential limitations of the information-switching module, such as overfitting risks or reduced performance with fewer modalities. Additional experiments with fewer modalities or reduced data could strengthen this aspect.

4.  While the paper discusses future work on model compression and optimization, it does not provide any preliminary results or insights into how these challenges might be addressed, leaving practical deployment concerns unresolved. 

5. The interpretability analysis using Grad-CAM is compelling but could be further supported by quantitative evaluations or feedback from clinical experts to validate the clinical utility of the highlighted regions. Besides, some related works about enhancing the interpretability of medical imaging analysis should be discussed [2 ].

[2] Li, C., Liu, X., Li, W., Wang, C., Liu, H., Liu, Y., ... & Yuan, Y. (2024). U-kan makes strong backbone for medical image segmentation and generation. arXiv preprint arXiv:2406.02918.

6.  The ablation studies focus mainly on specific components of the model. There is limited evaluation of the sensitivity of results to dataset size, noise, or variations in imaging quality, which could impact real-world performance.

Author Response

Comments 1: The abstract is too long and is hard to follow. Besides, there is a significant typo error in Line 358 with the funny sentence "acne. Error! Reference source not found."

Response 1: Thank you for pointing this out. We agree with this comment. Therefore, we have revised the abstract according to the journal's format and streamlined it to make it more concise and easier to follow. The word count of the abstract has been reduced. The revised abstract is marked in red.

Comments 2: The paper lacks sufficient implementation details, such as specific hyperparameter values, model configurations, and training runtime, which may hinder reproducibility by other researchers.

Response 2: We agree with this comment. Therefore, we have added Experimental configuration details of the MDSIS-Net model in Section 3.6, Table 4, which includes specific hyperparameter values, model configurations, and training runtime to enhance reproducibility for other researchers.

Comments 3: Although the proposed model outperforms baselines, the paper does not provide a detailed discussion or comparison with other state-of-the-art multi-modal approaches [1], especially those employing recently advanced transformer architectures.

[1] Jie, Y., Li, X., Tan, H., Zhou, F., & Wang, G. (2024). Multi-modal medical image fusion via multi-dictionary and truncated Huber filtering. Biomedical Signal Processing and Control, 88, 105671.

Response 3: Thank you for pointing this out. We agree with this comment. To address this, we have conducted additional comparative experiments with the state-of-the-art multi-modal transformer-based approach [1]. In Table 5 and Table 6, we have included detailed performance metrics comparing our proposed model with the advanced transformer-based model on both the DD and MM datasets.

  • Jie, Y., Li, X., Tan, H., Zhou, F., & Wang, G. (2024). Multi-modal medical image fusion via multi-dictionary and truncated Huber filtering. Biomedical Signal Processing and Control, 88, 105671.

Comments 4: The paper does not discuss potential limitations of the information-switching module, such as overfitting risks or reduced performance with fewer modalities. Additional experiments with fewer modalities or reduced data could strengthen this aspect.

Response 4: Thank you for pointing this out. We agree with this comment. Thank you for pointing this out. We agree with this comment. We have added experiments in Sections: Performance with fewer modalities, and Performance with shifting data distributions in the Supplementary information. Table S5 presents the model's performance under different numbers of modalities. These results highlight the importance of incorporating multiple modalities to achieve better performance. Furthermore, Table S6 shows the model's performance under different data splits. These results highlight the sensitivity of the model to the distribution of data across training, validation, and testing sets, and demonstrate that increasing the amount of training data can enhance the model's performance.

Comments 5: While the paper discusses future work on model compression and optimization, it does not provide any preliminary results or insights into how these challenges might be addressed, leaving practical deployment concerns unresolved.

Response 5: Thank you for pointing this out. We agree with this comment. We have conducted initial tests using Automatic Mixed Precision (AMP) training, which switches from float32 to float16 precision. These tests show promise for optimization, as they reduce memory consumption by half while incurring only a 1% accuracy loss. We have also added this content to the Discussion section, providing a direction for future work on compression and computation optimization.

Comments 6: The interpretability analysis using Grad-CAM is compelling but could be further supported by quantitative evaluations or feedback from clinical experts to validate the clinical utility of the highlighted regions. Besides, some related works about enhancing the interpretability of medical imaging analysis should be discussed [2 ].

  • Li, C., Liu, X., Li, W., Wang, C., Liu, H., Liu, Y., ... & Yuan, Y. (2024). U-kan makes strong backbone for medical image segmentation and generation. arXiv preprint arXiv:2406.02918.

Response 6: Thank you for pointing this out. We agree with this comment. Thank you for pointing this out. We revise all Grad-CAM visualizations as suggested. Specifically, for the Grad-CAM heatmaps generated by our proposed model, we apply the findContours algorithm based on heatmap intensity to generate lesion edges (shown by the blue boundaries). Additionally, we include the ground truth lesion regions of interest (ROIs) (shown by the red bounding boxes) annotated by dermatologists. These modifications significantly enhance the interpretability of medical imaging analysis.

Comments 7: The ablation studies focus mainly on specific components of the model. There is limited evaluation of the sensitivity of results to dataset size, noise, or variations in imaging quality, which could impact real-world performance.

Response 7: Thank you for pointing this out. We agree with this comment. We have added experiments on the Section: Model sensitivity to data variations in the Supplementary information. We conduct comparisons under different data variations, including changes in input size, blur, and the presence of noise. The results, listed in Table S4, demonstrate the model's performance under these variations. These findings confirm that since image-based clinical diagnosis places high demands on image quality, performance slightly declines with average blur and Gaussian noise. This indicates that while the model is robust to common data variations, it can still be affected by poor image quality.

Reviewer 2 Report

Comments and Suggestions for Authors

Authors presented a multimodal artificial neural network MDSIS-Net for recognizing skin lesions and applied it to classify lesions in several classes in deforming dermatosis and malignant melanoma. The model implements convolutional neural networks and a cross-attention mechanism to exchange information between different skin image modalities. The model exhibits a high classification accuracy compared to existing methods. The authors also use Grad-CAM as a visualization method to analyze the interpretability of the classification results, with the hope that clinicians may use these results to better understand the diagnostic results. Overall, the study looks solid, and the presented model seems promising for future applications and further improvement.

I have the following comments and questions:

-       The text is too lengthy. I recommend reformatting the manuscript to make it shorter and more concise, moving most of the technical details to the supporting information. This will definitely improve the clarity of the material for the wider audience. In particular, the abstract must be shortened by at least half. Section 4.2 can be moved to supporting information, with the main conclusions (without figures and lengthy description for each considered case) presented as a paragraph in the main text.

-       There are no sufficient details about the datasets used to train and test the model. How exactly was the data annotated? What criteria were used for labeling? The MM dataset is highly imbalanced (according to Fig. 17b — and it’s the first place where we find this out), so how was this handled?

-       I see how the feature maps visualized by Grad-CAM can help to understand which part of the image is important for classification, but it is not entirely clear how exactly these visualizations help clinicians in their decision making. It would be useful to include examples of clinical cases where Grad-CAM visualization helped in diagnosis.

-       The authors should discuss potential limitations of the model, such as its applicability to other types of skin diseases or its robustness to noise in the data.

-       Line 182 and similar: The links to figures and tables are completely broken in the text.

-       Lines 552–558 and section 4.1.1 should be a part of Methods, as they do not contain any results.

-       Tables 3 and 4: Were these results obtained on a testing data set (please indicate this in the captions)? As the numbers in the tables are close to each other, what is the statistical significance of the improvement stated by the authors?

-       Comparing Figures 8 and 10: Your model outperforms more clearly for MM as compared to DD. Are there any explanations?

-       Figure 11 and similar figures: What is the color coding for the Grad-CAM images, i.e. what should I look at in these images to see the important ‘feature maps’? I also see that lips are important in some Grad-CAM images in Figure 11 (if I guessed the color coding correctly), isn’t it a wrong feature extracted by the model?

-       Figure 14: An artificial object is present in the figure marked as the significant feature maps, isn’t it a wrong behavior of the model?

-       Lines 713–714: Please explain the “demonstrated efficiency in distinguishing features of MM and NMSC,” if both classes are completely mixed in Figure 16b.

-       As I see, part of the clinical data was new in this study (not published before), but the manuscript does not discuss ethical issues related to the use of this data.

Author Response

Comments 1: The text is too lengthy. I recommend reformatting the manuscript to make it shorter and more concise, moving most of the technical details to the supporting information. This will definitely improve the clarity of the material for the wider audience. In particular, the abstract must be shortened by at least half. Section 4.2 can be moved to supporting information, with the main conclusions (without figures and lengthy description for each considered case) presented as a paragraph in the main text.

Response 1: Thank you for pointing this out. We agree with this comment. We have revised the abstract according to the journal's format and streamlined it to make it more concise and easier to follow. The word count of the abstract has been significantly reduced. Additionally, we have streamlined the Introduction section and moved Section 4.2 “Ablation study” to the Supporting Information. These changes make the manuscript shorter, as most of the technical details are now in the Supporting Information, thereby improving the clarity of the material for a wider audience.

Comments 2: There are no sufficient details about the datasets used to train and test the model. How exactly was the data annotated? What criteria were used for labeling? The MM dataset is highly imbalanced (according to Fig. 17b — and it’s the first place where we find this out), so how was this handled?

Response 2: Thank you for pointing this out. We agree with this comment. To address these concerns, we have added Table 1: Data distribution and purpose for the DD and MM Datasets in Section 3.1, which provides sufficient details about the datasets used for training, validation, and testing of the model. Specifically, the clinical labels for the VISIA dataset are provided by experienced dermatologists, who assign diagnostic labels based on the lesion responses observed across different modalities. Additionally, in Section 3.2.2: Multi-modal Image Augmentation, we have included details on how image augmentation techniques are employed to mitigate class imbalance and enhance data diversity. These techniques include random rotations, flips, and color jittering, which help balance the dataset and improve the model's robustness.

Comments 3:  I see how the feature maps visualized by Grad-CAM can help to understand which part of the image is important for classification, but it is not entirely clear how exactly these visualizations help clinicians in their decision making. It would be useful to include examples of clinical cases where Grad-CAM visualization helped in diagnosis.

Response 3: Thank you for pointing this out. We agree with this comment. We revise all Grad-CAM visualizations as suggested. Specifically, for the Grad-CAM heatmaps generated by our proposed model, we apply the findContours algorithm based on heatmap intensity to generate lesion edges (shown by the blue boundaries). Additionally, we include the ground truth lesion regions of interest (ROIs) (shown by the red bounding boxes) annotated by dermatologists. These modifications significantly enhance the interpretability of medical imaging analysis and provide a clear visual aid to assist clinicians in identifying and diagnosing lesions more accurately.

Comments 4:  The authors should discuss potential limitations of the model, such as its applicability to other types of skin diseases or its robustness to noise in the data.

Response 4: Thank you for pointing this out. We agree with this comment. In the Discussion section, we discuss the potential limitations of the model, including its applicability to other types of skin diseases and its robustness to noise in the data. We have added experiments in Sections: Performance with fewer modalities, and Performance with shifting data distributions in the Supplementary information. We conduct comparisons under different data variations, including changes in input size, blur, and the presence of noise. The results, listed in Table S4, demonstrate the model's performance under these variations. These findings confirm that since image-based clinical diagnosis places high demands on image quality, performance slightly declines with average blur and Gaussian noise. This indicates that while the model is robust to common data variations, it can still be affected by poor image quality. Additionally, Table S5 presents the model's performance under different numbers of modalities. These results highlight the importance of incorporating multiple modalities to achieve better performance. Furthermore, Table S6 shows the model's performance under different data splits. These results highlight the sensitivity of the model to the distribution of data across training, validation, and testing sets, and demonstrate that increasing the amount of training data can enhance the model's performance.

Comments 5: Line 182 and similar: The links to figures and tables are completely broken in the text.

Response 5: Thank you for pointing this out. We agree with this comment. We have revised the cross-referencing of figures and tables throughout the manuscript to ensure all links are correctly formatted and functional.

Comments 6: Lines 552–558 and section 4.1.1 should be a part of Methods, as they do not contain any results.

Response 6: Thank you for pointing this out. We agree with this comment. We have moved this section to the Methods part.

Comments 7: Tables 3 and 4: Were these results obtained on a testing data set (please indicate this in the captions)? As the numbers in the tables are close to each other, what is the statistical significance of the improvement stated by the authors?

Response 7: Thank you for pointing this out. We agree with this comment. We have updated the captions of these two tables to explicitly indicate that the results are based on the testing dataset. Additionally, we have included p-values in both tables, calculated using a paired t-test, to demonstrate the statistical significance of the improvements achieved by the proposed model.

Comments 8: Comparing Figures 8 and 10: Your model outperforms more clearly for MM as compared to DD. Are there any explanations?

Response 8: Thank you for pointing this out. The performance difference can be attributed to the fact that the DD dataset is larger and more balanced in terms of class distribution compared to the MM dataset, which contributes to relatively better performance on DD.

Comments 9: Figure 11 and similar figures: What is the color coding for the Grad-CAM images, i.e. what should I look at in these images to see the important ‘feature maps’? I also see that lips are important in some Grad-CAM images in Figure 11 (if I guessed the color coding correctly), isn’t it a wrong feature extracted by the model?

Response 9: Thank you for pointing this out. We agree with this comment. For the "red marks", we use a 90-255 color coding for the Grad-CAM images, where higher intensity values indicate regions of greater importance. In the Interpretability analysis of the multi-modal skin lesion section, we have added a paragraph explaining that in some cases, such as the "red marks" on lips, the heatmap may exhibit misidentifications due to the similarity in color and texture between healthy lip regions and dermatitis-affected areas. This issue may arise because our lesion area segmentation is based on an unsupervised method.

Comments 10: Figure 14: An artificial object is present in the figure marked as the significant feature maps, isn’t it a wrong behavior of the model?

Response 10: Thank you for pointing this out. We agree with this comment. We have expanded the explanation for Figure 14 as follows: It is worth noting that an artificial object in Figure 14 is marked as a significant feature in the heatmap, which may be attributed to the similarity in visual patterns between the object and MM characteristics. Despite this, the heatmap analysis closely aligns with clinical assessments conducted by dermatologists. This can aid dermatologists in making more accurate diagnoses, optimizing treatment strategies, and improving care for patients managing MM.

Comments 11: Lines 713–714: Please explain the “demonstrated efficiency in distinguishing features of MM and NMSC,” if both classes are completely mixed in Figure 16b

Response 11: Thank you for pointing this out. We agree with this comment. We have added an explanation for Figure 16b: Our proposed model has demonstrated efficiency in distinguishing features of MM and NMSC, as depicted in the t-SNE plot in Figure 16b, where MM features tend to cluster in the inner layer, while NMSC features are distributed around the outer layer of MM. Additionally, we have modified the t-SNE visualization layer in the code, using the logits layer for visualization, and have updated all MM t-SNE visualization results throughout the manuscript accordingly.

Comments 12: As I see, part of the clinical data was new in this study (not published before), but the manuscript does not discuss ethical issues related to the use of this data.

Response 12: Thank you for pointing this out. We agree with this comment. In the Multi-modal skin lesion dataset section, we have added a description of the clinical data ethical considerations. The study was approved by the Ethics Committee of the First Affiliated Hospital of Ningbo University on December 20, 2023 (approval No. 2023R-178RS).

Reviewer 3 Report

Comments and Suggestions for Authors

The paper by Yu et al describes application of deep-learning techniques for the analysis of skin images. The authors demonstrated high accuracy for different skin lesions identification. Theme of the study is important and interesting, however, the presentation of the material in the manuscript requires additional improvements.

1. Abstract format should be done according to journal gudelines.

2. Many abbreviations in the abstract are not explained.

3. Line 182: "Error! Reference source not found." The same error appears in multiple lines, please check it carefully.

4. Introduction is too long.

5.  Equations 16-20 belongs to methods.

6. Term "steps" should be replaced with "epochs".

7. Discussion section is missing.

8. Overall length of the manuscript is too high. Please, shift part of the information into supplementary file.

9. The authors state " Every dataset is separated into testing, validation, and training sets." It will help a lot if the authors will clearly demonstrate in the Results section performance for  testing, validation, and training sets.

10. How the split into testing, validation, and training sets was performed? What will happen if some portion of data will be shift between these sets?

The paper may be published after correction of mentioned issues.

Author Response

Comments 1: Abstract format should be done according to journal gudelines.

Response 1: Thank you for pointing this out. We agree with this comment. We have revised the abstract according to the journal's format and streamlined it to make it more concise and easier to follow. The word count of the abstract has been significantly reduced. Additionally, we have streamlined the Introduction section and moved Section 4.2 “Ablation study” to the Supporting Information. These changes make the manuscript shorter, as most of the technical details are now in the Supporting Information, thereby improving the clarity of the material for a wider audience.

Comments 2: Many abbreviations in the abstract are not explained.

Response 2: Thank you for pointing this out. We agree with this comment. We have added explanations for abbreviations used in the abstract to ensure clarity and readability.

Comments 3: Line 182: "Error! Reference source not found." The same error appears in multiple lines, please check it carefully.

Response 3: Thank you for pointing this out. We agree with this comment. We have revised the cross-referencing of figures and tables throughout the manuscript to ensure all links are correctly formatted and functional.

Comments 4: Introduction is too long.

Response 4: Thank you for pointing this out. We agree with this comment. We have streamlined the Introduction section.

Comments 5: Equations 16-20 belongs to methods.

Response 5: Thank you for pointing this out. We agree with this comment. We have moved Equations 16-20 to the Methods part.

Comments 6: Term "steps" should be replaced with "epochs".

Response 6: Thank you for pointing this out. We agree with this comment. We have replaced the term "steps" with "epochs".

Comments 7: Discussion section is missing.

Response 7: Thank you for pointing this out. We agree with this comment. We have added the discussion section.

Comments 8: Overall length of the manuscript is too high. Please, shift part of the information into supplementary file.

Response 8: Thank you for pointing this out. We agree with this comment. We have streamlined the Introduction section and moved Section 4.2 “Ablation study” to the Supporting Information. These changes make the manuscript shorter, as most of the technical details are now in the Supporting Information, thereby improving the clarity of the material for a wider audience.

Comments 9: The authors state " Every dataset is separated into testing, validation, and training sets." It will help a lot if the authors will clearly demonstrate in the Results section performance for testing, validation, and training sets.

Response 9: Thank you for pointing this out. We have added Table 1: Data distribution and purpose for the DD and MM Datasets in Section 3.1: Multi-modal Skin Lesion Dataset, which provides sufficient details about the datasets used for training, validation, and testing of the model. The training set is used for training our deep learning model, the validation set is used to select the best-performing model, which is then evaluated on the test set, and the test set is used to evaluate the performance of the model. All performance metrics in this paper are calculated on the test set.

Comments 10: How the split into testing, validation, and training sets was performed? What will happen if some portion of data will be shift between these sets?

Response 10: Thank you for pointing this out. We have added experiments on Sections: Performance with shifting data distributions in the Supplementary information. Table S6 shows the model's performance under different data splits. These results highlight the sensitivity of the model to the distribution of data across training, validation, and testing sets, and demonstrate that increasing the amount of training data can enhance the model's performance.

Round 2

Reviewer 1 Report

Comments and Suggestions for Authors

The revision has addressed my concerns.

Author Response

Thank you.

Reviewer 2 Report

Comments and Suggestions for Authors

I appreciate the efforts made by the authors to improve the manuscript.

Author Response

Thank you.

Reviewer 3 Report

Comments and Suggestions for Authors

The authors improved the manuscript presentation and readability, but still there are major issues:

  1. There is no strict comparison of the proposed models perfomances for training, validation and test.
  2. Discussion section is missing references to the studies in the field.
  3. Reference to supplementary materials must be presented after conclusion. Reference to supplementary may be given rirht in the results.
  4. Number of illustrations in my opinion is still too large. I propose to combine some images or shift some of them to supplementary file.

The ppaer may be published after correction of mentioned issues.

Author Response

Comments 1: There is no strict comparison of the proposed models perfomances for training, validation and test.

Response 1: Thank you for pointing this out. We agree with this comment. We have added results in Sections: Performance across training, validation, and test sets in the Supplementary information. The model's performance is evaluated across training, validation, and test sets, as shown in Table S7.

Comments 2: Discussion section is missing references to the studies in the field.

Response 2: Thank you for pointing this out. We agree with this comment. We have addressed the issue by adding 8 relevant references to the Discussion section in this field.

Comments 3: Reference to supplementary materials must be presented after conclusion. Reference to supplementary may be given rirht in the results.

Response 3: Thank you for pointing this out. We agree with this comment. We have moved the reference to the supplementary materials to the section after the conclusion, as recommended.

Comments 4: Number of illustrations in my opinion is still too large. I propose to combine some images or shift some of them to supplementary file.

Response 4: Thank you for pointing this out. We agree with this comment. We have moved four figures from the Methods section and two figures from the Result section to the supplementary file. This adjustment helps streamline the main text while ensuring that all relevant visual materials remain accessible.

Round 3

Reviewer 3 Report

Comments and Suggestions for Authors

The authors addressed arised issues, the paper may be published.